Estimated effects of implementing an open access policy for grantees at a private foundation

Strasser Carly carlystrasser@gmail.com 1
Khare Eesha 2
1 Gordon and Betty Moore Foundation , Palo Alto , CA , United States of America
2 Harvard University , Cambridge , MA , United States of America
Marusic Ana
Electronic publication date: 2017 Sep 26
Publication date: 2017
Volume: 5
Electronic Location ID: e3853
Received 2017 May 10; Accepted 2017 Sep 5
Copyright: ©2017 Strasser and Khare
Copyright year: 2017
Copyright holder: Strasser and Khare
License: This is an open access article distributed under the terms of the Creative Commons Attribution License, which permits unrestricted use, distribution, reproduction and adaptation in any medium and for any purpose provided that it is properly attributed. For attribution, the original author(s), title, publication source (PeerJ) and either DOI or URL of the article must be cited.
License URL: https://creativecommons.org/licenses/by/4.0/

Keywords: Open access, Policy, Philanthropy

Funding: Gordon and Betty Moore Foundation This work was supported by the Gordon and Betty Moore Foundation through Strasser’s in-kind contributions and support for Khare as an intern. The funders had no role in study design, data collection and analysis, decision to publish, or preparation of the manuscript.

==============================
Background

The Gordon and Betty Moore Foundation (GBMF) was interested in understanding the potential effects of requiring that grantees publish their peer-reviewed research in open access journals.

Methods

We collected data on more than 2,000 publications in over 500 journals that were generated by GBMF grantees since 2001. We then examined the journal policies to establish how two possible open access policies might have affected grantee publishing habits.

Results

We found that 99.3% of the articles published by grantees would have complied with a policy that requires open access within 12 months of publication. We also estimated the maximum annual costs to GBMF for covering fees associated with “gold open access” to be between $400,000 and $2,600,000 annually.

Discussion

Based in part on this study, GBMF has implemented a new open access policy that requires grantees make peer-reviewed publications fully available within 12 months.

Introduction

In recent years, evidence of the benefits of open access (OA) for stakeholders in scientific research has been mounting. OA benefits researchers by increasing visibility of their research (Wang et al., 2015) and increasing citations counts (Gargouri et al., 2010). OA also helps build research capacity in developing countries (Chan, Kirsop & Arunachalam, 2005), and decreases financial pressure on academic and research libraries (McGuigan & Russell, 2008). Funders are interested in promoting OA to increase visibility of the work that they fund, thereby increasing its potential impact. Increased visibility of research also reduces duplicative efforts and therefore duplicative investment. Further, OA research is available not just to academic researchers, but also to industry and the general public, thereby increasing reach and potentially resulting in a higher return on investment (Tennant et al., 2016). Because of the growing body of evidence around OA benefits, many funders are establishing OA policies that mandate unrestricted online access to articles published in scholarly journals. Funders who have implemented policies recently include the Bill and Melinda Gates Foundation, the European Commission, US federal funders including National Science Foundation and the National Institutes of Health, and the Wellcome Trust.

It is generally agreed that “open access” refers to online research outputs that are free of all restrictions on access (e.g., fees or login requirements) and free of many restrictions on use (e.g., certain copyright and license restrictions) (Suber, 2012). Here we consider the two main ways that authors can make their work OA, gold and green (Harnad et al., 2008). In gold OA, the content is freely accessible at the time of publication. Some journals require article processing charges (APCs) to cover the cost of making content freely available. Green OA refers to author “self-archiving”, i.e., the content is posted online in an institutional or subject repository, or to a personal website. This method requires compliance with publisher or journal policies on self-archiving, and are often subject to embargoes (i.e., a period during which an OA version of the article cannot be made public via self-archiving).

There is evidence that researchers are starting to consider openness in journal selection (Priem, 2013), however this shift has yet to result in most articles being OA (Piwowar et al., 2017). Anecdotal evidence suggests most authors may not select journals based on their level of openness or their policies around self-archiving, instead choosing the most relevant, highest impact journal for their field. This is further complicated by the fact that some of the journals with highest impact factor are closed and do not allow for green or gold OA options. Funder policies around publishing OA would require that researchers more carefully consider their choice of journal to ensure compliance.

In an effort to increase access to the research results it funds, in 2016 the Gordon and Betty Moore Foundation (GBMF) began considering the implementation of an open access policy for all publications produced by its grantees. GBMF funds research in basic science, environmental conservation, patient care improvements, and preservation of the San Francisco Bay Area. With an annual budget of approximately $300 million, GBMF has awarded approximately 2,400 grants since its founding in 2001. Although the foundation’s existing Data Sharing and Intellectual Property Policy generally favors public access to grant outputs, at the time of this study it did not mandate open access.

Prior to implementing a new policy mandating open access, GBMF was interested in better understanding its effects on both grantees and the foundation’s spending. GBMF was particularly interested in ensuring any policy did not infringe on the independence and expertise of grantees. That is, GBMF operates on the principle that grantees are most attuned to the needs of their discipline, and most likely to understand how best to disseminate their research to ensure high impact. GBMF carefully considers any policies that might restrict the normal activities of its grantees regarding their career path, scholarship, or results dissemination. An OA policy prohibits grantees from publishing in the journal they believe is best would not be ideal since it assumes we are better informed as to how the grantee should conduct their research endeavors. The current policy of the foundation encourages openness, but does not weigh in on journal choice.

We explored the potential costs of a policy change for GBMF that mandated OA, and how this policy may affect journal choice for various types of GBMF grantees. We were most interested in (1) whether grantees would be restricted from publishing in journals they have previously chosen, and (2) the financial ramifications of a policy advocating for and funding gold OA when available. Journals chosen by grantees in the past were not influenced by any GBMF requirements or policies; we may assume that the list of journals we report here may be influenced in the future by such policies. Since this work was completed, GBMF has implemented a new open access policy that requires all peer-reviewed publications be openly available within 12 months of publishing.

Although many funding organizations have mandated OA in some form, we are not aware of any that have published results from internal analyses on the effects of such mandates. Kiley (2014) provided data on Wellcome Trust’s spending on APCs for grantees, however this was not analyzed formally in a publicly available article. The dearth of available information about effects of OA from a funder perspective is at least in part due to eclectic methods of grantee reporting used by different funders. Grantee reports are not often machine readable or easily analyzed in bulk. Some funders employ external firms to collect data on grantees’ activities and publications, but these are primarily used for internal decision-making and are not made public. Other funders interested in pursuing mandates may benefit from understanding the results of our study, and the transparency of publishing our analyses is evidence of our commitment to promoting open research. Other potential audiences include policy makers and scholarly communication researchers.

Methods

To explore potential impacts of an open access policy at GBMF, we analyzed 2,650 publications produced by GBMF grantees between 2001 and 2017 (Strasser & Khare, 2017a). This is not a complete list of publications since the foundation does not yet have a standardized way of collecting grantee outputs. The dataset includes publication data obtained from Science Program grantee reports, as well as publication data from Crossref’s Funding Data Search service (Crossref, 2017). Publications were deduplicated, grouped by journal title, and journal policy metadata was added by searching the SHERPA-RoMEO database (Jisc, 2017) of publisher policies on self-archiving and open access.

Based on information found on journal websites and SHERPA-RoMEO, we classified each journal as either open access, closed access, or hybrid. OA journals provide access to all content immediately online (gold OA). Closed access journals restrict access to their content by requiring that readers log into their website, usually to verify access to an institutional subscription. Hybrid journals are closed access journals that provide authors with the option to opt into OA by paying a fee (gold OA); only those articles that are designated OA are available publicly online. We also determined whether the journal allowed authors to archive post-prints (peer-reviewed versions of articles, but not necessarily with the publisher’s formatting)—known as green OA, and the length of the embargo period for archiving post-prints. The number of articles per journal is also included in this article’s corresponding dataset (Strasser & Khare, 2017b).

The potential costs to GBMF for gold OA publishing were of interest, regardless of the policy chosen. This would be particularly relevant if GBMF planned to provide financial support for grantees publishing gold OA in journals with APCs. We therefore calculated an annual estimate of maximum costs for gold OA for 2009–2016. We chose to analyze only a subset of years since our dataset is limited and we have reason to assume the number of publications in years prior to 2009 are vastly undercounted (less than 100 publications were found in years prior to 2009 in our data collection). We multiplied the number of articles published in a year by 0.89, which is the percent of articles from our dataset that were published in hybrid or open journals and could therefore be made gold OA. We then estimated the maximum annual cost for making these articles OA by multiplying the value by $3,000, the high end of the APC estimate from West, Bergstrom & Bergstrom (2014). Others report lower average APC costs (e.g., Romeu et al., 2014), however we chose the highest estimate to capture the fact that this was intended to estimate the potential maximum cost of OA.

Our study is limited by the availability of consistent, complete data for GBMF grantees. Annual reports are not collected in consistent, machine readable ways that lend themselves to bulk analysis. We also have no data on APCs already paid for using GBMF funds out of grantee budgets. Efforts are underway at GBMF to develop systems for consistent, reliable collection of grantee publication data, however these were not implemented in time to benefit this study.

Results

Our data collection yielded a list of 573 journals used by grantees, in which 2,650 articles were published. We were first interested in the percentage of journals chosen by our grantees that are hybrid, open, or closed. We calculated percentages both by journal and by article to ensure that we captured potential effects on authors, however the numbers were quite similar (Fig. 1). GBMF grantees tend to publish in hybrid journals (74% of journals; 72% of articles), with open (16% of journals; 17% of articles) and closed (10% of journals; 11% of articles) journals less represented. Although we have no information on the percentage of those articles published in hybrid journals that were published gold OA, there is evidence that without funder mandates, the percentage of articles that are published OA is low (Tennant et al., 2016).

Figure 1 Breakdown of journal type (open, hybrid, or closed) used by GBMF grantees for the 2,650 articles (A) published in 573 journals (B).

Data: Strasser & Khare (2017a) and Strasser & Khare (2017b).

We were also interested in whether a grantee’s journal choice would be impacted by two possible OA policies being considered. Policy A would require OA within 12 months of publication, either via the green or gold route. Grantees could comply by either publishing in open journals, by publishing OA in a hybrid journal (gold OA), or by publishing non-OA and self-archiving within 12 months (green OA). This policy would preclude grantees from publishing in journals that are closed and restrict self-archiving for longer than 12 months. Policy B would require immediate access at the time of publication (gold OA). Grantees could comply by either publishing in open journals, or by publishing OA in a hybrid journal. In some cases, journals may allow immediate access at the time of publication (no embargoes on post-prints); this route of OA would also be acceptable (but is much less frequent in journals used by grantees). Policy B excludes journals that are closed and/or restrict self-archiving at the time of publication.

Based on these proposed policies, we calculated that the percentage of articles in our dataset that would have been compatible with Policy A is 99.3. The articles that would not have been compatible with Policy A were exclusively from the family of Annual Reviews. These journals are not typical publications since they do not publish original research and are instead reviews of work in various disciplines.

For Policy B, 8.3% of articles in our dataset would not have been compatible. These are primarily high impact journals with no gold OA option that restrict post-print archiving for some period of time. Journals that do not comply include the family of Nature journals (except Nature Communications) and Science, published by AAAS (American Association for the Advancement of Science).

The estimated maximum cost for implementing Policy B (i.e., gold OA for all publications) ranged from $401,835 in 2008 to $2,538,725 in 2015. (Fig. 2). This large range is a direct consequence of the variable amount of data available on articles across years. For example, we identified 814 publications in 2015 (compared to 328 and 501 in the years before and after, respectively). In 2015, the newly funded Data-Driven Discovery Initiative (https://www.moore.org/initiative-strategy-detail?initiativeId=data-driven-discovery) in the Science Program began systematically collecting data from its grantees about publications, resulting in the large spike. Such inconsistencies in the data are unavoidable given the absence of methods for collecting data systematically.

Figure 2 Estimated maximum annual costs for OA fees associated with GBMF grantee publications.

Data: Strasser & Khare (2017a) and Strasser & Khare (2017b).

Discussion

Based on the data reported here, the OA policies considered by GBMF would not have substantially altered the journals in which the grantees published their work. This is particularly true for proposed Policy A (green or gold OA within 12 months). Only 0.7% of articles published by grantees would not have complied with this policy. If the gold OA policy option were enacted (Policy B), some journals that restrict access to all their content for up to 12 months after publication would not be permissible. Only 8.3% of articles published by grantees in our dataset would not have complied with this policy.

Implementing an OA policy at a private foundation has potential implications beyond ensuring grantees make their work open access. It also serves to advocate for a position of openness in research outputs. Such a policy might encourage grantees to select journals with more open policies, or may encourage them to use grant funds to choose gold OA (even without a mandate). Of course, requirements imposed by funders for OA will inevitably result in more accessible work. A study by Xia et al. (2012) found that hundreds of policies have been proposed and adopted at various organizational levels and many of them have resulted in increased self-archiving (i.e., green OA).

The potential maximum financial ramifications of paying APCs for all articles were estimated to be between $400,000 and $2,600,000 per year. There are several unknown factors that might influence what amount within this range is likely to be correct for GBMF, most critically how many publications are generated each year by grantees. Our estimates of number of publications per year (and therefore estimated maximum APC costs) rely on either the grantee self-reporting to the foundation, or the grantee including GBMF in manuscript acknowledgements that can be harvested by Crossref. Our range of costs is therefore likely an underestimate for earlier years that have lower publication counts due to the increased difficulty in tracking down data from those years. Potential overestimates at the higher end of the range would result from assuming GBMF would incur costs for all publications generated by grantees.

The estimated maximum cost for implementing Policy B (i.e., gold OA for all publications) is the result of several simplifications. First, we account for any potential institutional memberships, wherein publishers charge institutions a fixed annual fee which covers some percentage of article-processing fees for the institution for the year. These memberships may result in discounts, which may result in significantly reducing costs for OA (Smith & Sartori, 2017). Second, our calculation also does not correct for the fact that many OA journals do not charge APCs (see http://www.doaj.org), and therefore the maximum cost for GBMF to fund gold OA would be lower.

There are other considerations in OA policy implementation that were not the subject of this research. For instance, some funders have chosen to cease funding APCs for OA articles in hybrid journals, opting instead to use funds to cover costs for fully OA journals only. This strategy is used by the German Research Foundation and the Norwegian Research Council, and is a response to high prices and poor service, and some journals’ tendency to charge for articles where APCs have been paid due to internal error (Butler, 2016).

GBMF announced a new Open Access Policy in 2017 (Moore Foundation, 2017). The new policy states that

The foundation requires that a final (post-print) version of all peer-reviewed articles produced as a result of research supported, either in entirety or in part, by the foundation’s funding, be made publicly and freely available (open access, or OA) within 12 months of publication. Grantees can accomplish this either by publishing the article OA, by ensuring that the publisher will make the content OA within 12 months, or by depositing a post-print version of the manuscript in an OA repository within 12 months.

This is equivalent to Policy A described above. The decision to implement Policy A as opposed to something more prescriptive, such as Policy B, was based in part by GBMF’s policy to allow grantees to make decisions about their research and its dissemination based on their expertise, rather than our policies. GBMF funds a wide breadth of research, and not all grantees will be impacted by an OA policy in the same way. The Environmental Conservation Program and the Science Program both are likely to have grantees that produce peer-reviewed publications, and are likely to be impacted by the OA policy. However even within these two Programs, there are a diversity of disciplines represented that may require different amounts of behavioral changes in publishing habits to comply with the policy. The new policy at GBMF will serve to “level up” the different groups, ensuring that the public can access all peer-reviewed publications generated by its grantees.

Another factor in choosing Policy A over Policy B was the potential ramifications of encouraging gold OA without considering how fees will be paid. Some funders have separate budgets for APCs, while others require that this cost be included in the grant budget. GBMF was interested in a more conservative approach, which will result in time to gather data and concerns about implementing a policy that requires gold OA.

GBMF plans to revisit the efficacy and impact over time for the newly implemented policy. There are several variables that can be altered to potentially strengthen the OA policy. These may include (1) restricting embargoes on archiving OA versions to six months (compared to 12 months); (2) requiring CC-BY licenses for all publications; (3) setting aside funds for covering grantees’ OA fees (independent of their grant funds); (4) mandating particular repositories for OA archiving; or (5) expanding the policy to include outputs other than peer-reviewed journals (e.g., data, software, books, etc.).

The authors would like to thank N Caulk, J Lin, and A Jones for helpful comments on the manuscript. K Canesi provided data. C Mentzel and N. Caulk provided helpful insights throughout the project. We are especially grateful for the two reviewers (M Winker and anonymous) who gave helpful feedback that resulted in a much better manuscript.

Additional Information and Declarations

Competing Interests

Author Contributions

Data Availability

The authors declare there are no competing interests.

Carly Strasser conceived and designed the experiments, performed the experiments, analyzed the data, contributed reagents/materials/analysis tools, wrote the paper, prepared figures and/or tables, reviewed drafts of the paper.

Eesha Khare conceived and designed the experiments, performed the experiments, analyzed the data, contributed reagents/materials/analysis tools, wrote the paper, reviewed drafts of the paper.

The following information was supplied regarding data availability:

Strasser, Carly, & Khare, Eesha. (2017). Moore grantee publication data [Data set]. Zenodo. http://doi.org/10.5281/zenodo.555947.

Strasser, Carly, & Khare, Eesha. (2017). Moore grantee list of articles [Data set]. Zenodo. http://doi.org/10.5281/zenodo.841794.

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
