# Peer review of "Estimated effects of implementing an open access policy for grantees at a private foundation"

_PeerJ, doi:10.7717/peerj.3853_

## Round 0.1 · original submission · Major Revisions

· Academic Editor

Major Revisions

Please pay special attention to the comments of the Reviewer 2.

·

Basic reporting

1. The methodology is not very detailed (insufficient detail for replication), with some errors (see below).
2. A structured abstract would be preferable to make it clear this is considered research.
3. It would be helpful to have a basic description of the number of grants and the size of grants. The total amount of funding the GBMF provides annually should be provided, to provide a frame of reference.
4. “OA journals provide access to all content immediately online.” Most definitions of open access also include Creative Commons licenses, but they are not addressed until the last paragraph of the article and their benefits are not mentioned. If the authors are not considering CC as part of this evaluation, they should define up front that the type of OA being analyzed refers to free content only.
5. “We then estimated the annual cost for making these articles OA by multiplying by a low ($1500) and high ($3500) APC estimate (Solomon 2013).” This reference is not included in the reference list (and should be) so it wasn’t possible to verify that the estimates are reasonable and appropriate for the journals in the study.
6. “Instead, they are often more concerned with publishing in the most relevant, highest impact journal for their field (Priem 2013).” Priem 2013 doesn’t appear to say this—it is a thought piece about the future of how researchers will disseminate their work and indicates that researchers are already changing in how they do so. Is the reference wrong?
7. “To explore potential impacts of an open access policy at GBMF, we analyzed 2650 publications produced by GBMF grantees between 2001 and 2017” Did the choice of journal change over time?
8. “Figure 1. Breakdown of journal type (open, hybrid, or closed) used by GBMF grantees for the 2650 articles (left) published in 573 journals (right).” This notation of journal vs article is opposite what the figure shows. Which is correct?
9. Dataset: would be more transparent to review if Post Print Policy indicated Y for authors are allowed to publish and N for they are not allowed to (rather than 1 or 0).

Experimental design

1. The novelty and importance of such an analysis is not addressed. Have any other foundations published their analysis? Why is it important to share this information?
2. The design is not strong; the authors have no way of knowing how complete the data are; certainly the data in 2015 imply that the rest of the years are underestimated. The estimates vary 10-fold. The estimates for APC charges are simplistic and the reference for them is not listed.
3. The authors don’t actually provide the calculus of weighing pros and cons besides concluding that gold OA would be expensive (apparently – they don’t actually say that they concluded it would be too expensive for the foundation). The fact that gold OA would cost more than green OA is apparent without doing the analysis.

Validity of the findings

1. The authors do not know the number of papers actually published (the statement “We identified more than 800 publications in 2015 (compared to approximately 300 and 500 in the years before and after, respectively). In 2015, the newly funded Data-Driven Discovery Initiative in the Science Program began systematically collecting data from its grantees about publications, resulting in the large spike.” suggests that the articles in other years are substantially undercounted.
2. The gold PA cost estimates have a 10-fold range and are based on assumptions rather than actual charges; they acknowledge that the actual amounts may be lower or higher.
3. Factors such as institutional membership that would reduce expenses, or OA journals that do not charge APCs, are not taken into account.
4. The authors do not estimate the value of OA, and in fact only mention the benefits of OA in the introduction.
5. They don’t consider Creative Commons licenses.
6. Title, “Expected effects of an open access policy at a private foundation” This study does not address effects (including impact of the research if it is made freely available after a year, immediately, or never), only costs and journals that authors could publish in. Perhaps “Estimated costs of implementing an open access policy at a private foundation”
7. The authors state, ““OA offers a number of benefits including increased citations counts (Gargouri et al. 2010), accessibility for building research capacity in developing countries (Chan et al. 2005), enhanced visibility of research (Tennant et al. 2016), and decreased financial pressure on academic and research libraries (McGuian and Russell 2008).” They state, “In an effort to increase access to the research results it funds” but which specific benefits were of greatest importance for the GBMF? How does their new policy address these benefits?
8. Why did the GBMF feel it was important to let authors publish where they normally would rather than encouraging them to publish in OA journals
9. “grantee’s preferred journal” – it’s unknown what their preferred journal was; this was the journal they published in.
10. “GBMF plans to revisit the efficacy and impact over time for the newly implemented policy. There are several variables that can be altered to potentially strengthen the OA policy. These may include (1) restricting embargoes on archiving OA versions to six months (compared to 12 months); (2) requiring CC-BY licenses for all publications; (3) setting aside funds for covering grantees’ OA fees (independent of their grant funds); (4) mandating particular repositories for OA archiving; or (5) expanding the policy to include outputs other than peer-reviewed journals (e.g., data, software, books, etc.).” It is surprising to see this statement after the analysis. Why was the decision to use the green approach made and what factors would make GBMF rethink that decision?
11. “GBMF funds research in basic science, environmental conservation, patient care improvements, and preservation of the San Francisco Bay Area.” “Moore Foundation grantees tend to publish in hybrid journals (74% of journals; 72% of articles), with open (16% of journals; 17% of articles) and closed (10% of journals; 11% of articles) journals less represented.” The ratios of gold to green journals vary substantially by field (see Björk et al., 2010, doi: 10.1371/ journal.pone.0011273.g004 ). It would be useful to see whether the calculations vary substantially by field.
12. Cost estimates don’t take into account memberships (eg, In “Cambridge RCUK Block Grant spend for 2016-2017” https://unlockingresearch.blog.lib.cam.ac.uk/?p=1463 : “The average article processing charge was £1850 – this is significantly less than the £2008 average last year, reflecting the value of memberships”)

Additional comments

This paper provides the assessment conducted by a foundation in determining what OA policy to implement, and you should be lauded for being more transparent about your decision making than is often the case. However, the paper has a number of limitations. The limitations are in part due to the challenges of finding all the publications supported by GBMF and taking into account the actual APCs instead of the estimated APCs. However, if you don’t provide the caveats and frame the discussion appropriately, another foundation might look at this and make the same decision without digging deeper, which would be unfortunate. The decision-making process presumably involves weighing pros and cons, not simply (apparently) "that's a lot of money, we can't afford that." Therefore, while the transparency is laudable, the analysis is fairly simplistic and it doesn't really rise to the level of research. While a formal cost-benefit analysis is presumably more than you are willing to undertake, what proportion of grants would have to not be funded to make up for the expense of funding OA? This sort of estimate would be useful.

Reviewer 2 ·

Basic reporting

This paper presents an initiative of the private funding company for changes in their policy related to the open access to the articles written by the grantees. The study involves “more than 2000” publication in “more than 500 journals”, and approximate cost of OA, according two suggested OA policies, has been calculated and discussed.
There are wrong definition on gold and green OA present: “Gold Open Access” journals which are considered only as journals with APC business model. It is well known that many gold OA journals do not charge authors. Also "Green Open Access" is not related with the journals but with authors who deposit their works (not only journal articles) into different institutional, subject based or other type of OA repositories.
In the focus of the study are possible OA policies, but it is not clear why the funder wants to mandate OA. Benefits of OA for funders and researchers are not discussed. Funders have encouraged or mandated OA because enhanced visibility and discoverability increases the return of their investment, making the results of the funded research more widely available. Funder’s OA policy ensures the results of the research they fund can be read and used by anyone, including industry and society. Also, funder adopting OA policy avoids funding duplicative research, creates transparency, and encourages greater interaction with results of funded research. There are more benefits for researchers/grantees too. None of these are mentioned or discussed. There is only the phrase “potential implications” used in the discussion, but none “potential implication” is mentioned or discussed.
The manuscript was written in clear and professional English. Literature references, although not sufficient, are context provided. The missing literature includes mostly papers discussing OA mandates of other big funders. The manuscript is well structured and two figures are provided. Raw data is shared partially. Data on journals are included, but data on articles are missing.
The title is somehow misleading since no “expected effects of an open access” are discussed in the manuscript

Experimental design

Research questions are defined as an interest “in understanding (1) whether grantees would be restricted from publishing in their preferred journals, and (2) the financial ramifications of a policy advocating for and funding Gold OA when available.” Authors are not mentioning possible influences of the funders and the present research assessment criteria on the “preferred journals”, which could contribute to the study.
There is also kind of confusion with the years included in the study:
50 “In an effort to increase access to the research results it funds, The Gordon and Betty Moore Foundation (GBMF) began considering the implementation of an open access policy all publications produced by its grantees in 2016.”
65 “To explore potential impacts of an open access policy at GBMF, we analyzed 2650 publications produced by GBMF grantees between 2001 and 2017.”
76 “We also calculated an annual estimate of Gold OA costs for 2009-2016.”
Also, APC price ranges are not used consistently:
39 …on the order of $1000-3500 USD – USD needs to be deleted
later on, estimating the annual cost for making articles OA, authors are multiplying by a low ($1500) and high ($3500) – is it “low” $1000 or $1500??
Limitations of the study are not present.

Validity of the findings

Other minor changes include:
37 …OA are available publicly online.
77 …by 0.89 – what kind of data from Sherpa/Romeo was used to calculate this percent?
88 “Although we have no information on the percentage of those articles published in hybrid journals that were published Gold OA, we assume the percentage is low since this was not a requirement for grantees.” – information on OA articles by grantees could be easily retrieved, there is no need to estimate
102 “Based on these proposed policies, we calculated that the percentage of articles not compatible with Policy 1 is 0.7 - that is, 99.3% of articles would have been compatible with Policy 1.” – please leave not compatible, OR compatible part of the sentence.
104 two times “were”
105 Annual reviews are “explained” but there is no single reason present which can explain why they are close
120 “since these are the years where we have data for more than 100 articles.” – what kind of criteria is that??
121 in the Results expressions like “more than 800”, “approximately 300” and “approximately 500” cannot be used – please state accurate numbers

Additional comments

An interesting topic was presented that has been somehow. superficially elaborated.

---

## Round 0.2 · Minor Revisions

· Academic Editor

Minor Revisions

Thank you for your revised paper and answers to the reviewers comments. As you can see, there are some remaining comments which you need to address.

·

Basic reporting

The manuscript is substantially better reported and nearly all the original concerns have been met. There are a few points that should be clarified, below.

1. Title is better but still could be more clear, eg, "Estimated costs of implementing an open access publication policy for grantees of a private foundation"

2. Background: The title was revised but the Background is still overly broad. Perhaps: "The Gordon and Betty Moore Foundation (GBMF) was interested in understanding the potential costs of a policy requiring that grantees publish their peer-reviewed research in open access journals."

3. "Anecdotal evidence suggests most authors do not select journals based on their level of openness or their policies around self-archiving. Instead, they are often more concerned with publishing in, instead choosing the most relevant, highest impact journal for their field.": Without a reference the evidence is anecdotal, therefore the language should be more equivocal, eg, "anecdotal evidence suggests that authors may not select journals based on..."

4. "We also determined whether the journal allowed authors to archive post-prints (green OA)": "Post-prints" is not a commonly used term--"archive accepted or published manuscripts in a publicly accessible repository" would be more clear.

5. "Based on the data reported here, the OA policies considered by GBMF are unlikely to have significantly impacted the grantees’ choice of journal due to lack of compliance.":
"due to lack of compliance" is unclear -- maybe "would not have substantially altered the journals in which the grantees published their work" ?

Experimental design

The authors now adequately describe the limitations of the design in the Methods and Discussion.

Validity of the findings

The authors have addressed issues regarding impact and novelty, and corrected previous errors. Conclusions are appropriately circumspect.

Reviewer 2 ·

Basic reporting

no comment

Experimental design

BEFORE: “Although we have no information on the percentage of those articles published in hybrid journals that were published Gold OA, we assume the percentage is low since this was not a requirement for grantees.” – information on OA articles by grantees could be easily retrieved, there is no need to estimate
REVISED: „Although we have no information on the percentage of those articles published in hybrid journals that were published gold OA, there is evidence that without funder mandates, the percentage of articles that are published OA is low (Tennant et al. 2016).“
Authors' response: This work is beyond the scope of the project and would require a significant amount of additional research and work. It was not part of our original research questions. The authors are not aware of where this information can be “easily retrieved” in bulk.

Could not agree that this is „beyond the scope of the project“. It is very important from the funder perspective to know exactly what percentage of articles are already OA (without additional funder's investments in OA), and before the implementation of the OA policy (mandate). Since the hybrid journals are the most represented group (74%) such an evidence should be the starting point of this study. There is the way to do it „in bulk“ for the articles in HTML and PDF formats. If authors cannot do it „in bulk“ for various reasons, the data for the last two or three years (2014-2016) could be checked manually and provided.

„We were also interested in whether a grantee’s journal choice would be impacted by two possible OA policies being considered. Policy 1 would require OA within 12 months of publication, either via the green or gold route. Grantees could comply by either publishing in open journals, by publishing OA in a hybrid journal (gold OA), or by publishing non-OA and self-archiving within 12 months (green OA). This policy would preclude grantees from publishing in journals that are closed and restrict self-archiving for longer than 12 months. Policy 2 would require immediate access at the time of publication (gold OA). Grantees could comply by either publishing in open journals, or by publishing OA in a hybrid journal. This excludes journals that are closed and/or restrict self-archiving at the time of publication.” - The only difference between Policy 1 and 2 is the 12 month embargo period which reflects only on articles from the closed journals and the green OA (self-archiving) share. Green OA is free of charge, and there are no differences in costs for OA between Policy 1 and 2. Since there are different journal policies on self-archiving, some articles from the closed journals will be allowed for self-archiving immediately at the time of publication (or even before!), and some not. Both policies include gold and green route to OA. This is not clear from the descriptions of the Policies.

Validity of the findings

Abstract: “Based in part on this study, GBMF has implemented a new open access policy that requires grantees make peer-reviewed publications fully available within 12 months. – the authors did not consider the usage rights (Creative Commons licenses), although it would be necessary for such an analysis, and this should be corrected in „available for reading“.

The articles that would not have been compatible with Policy 1 were exclusively from the family of Annual Reviews. These journals are not typical publications since they do not publish original research; they are review series in specific disciplines in science and social science. – there are Annual reviews also in other disciplines (Annual Review of Applied Linguistics, Annual Review of Biomedical Sciences, Annual Review of Clinical Psychology, Annual Review of Energy, Annual Review of Genetics, Annual Review of Immunology, Annual Review of Neuroscience, Annual Review of Nutrition, Annual Review of Pharmacology and ToxicologyAnnual Review of Public Health, etc.)

„The estimated maximum cost for implementing Policy 2 (i.e., gold OA for all publications) ranged from $401,835 in 2008 to $2,538,725 in 2015. (Figure 2).“ – the estimated cost is valid for both policies since the only difference is embargo (and green OA respectively).

„For example, we identified 814 publications in 2015 (compared to 328 and 501 in the years before and after, respectively). In 2015, the newly funded Data-Driven Discovery Initiative (https://www.moore.org/initiative-strategy-detail?initiativeId=data-driven-discovery) in the Science Program began systematically collecting data from its grantees about publications, resulting in the large spike.“ – the given explanation does not explain the lower number of publications in 2016 when Data Driven Discovery Initiative was still in place.

“Our estimates of number of publications per year (and therefore estimated maximum APC costs) rely on either the grantee self-reporting to the foundation, or the grantee including GBMF in manuscript acknowledgements that can be harvested by Crossref. Our range of costs is therefore likely an underestimate for earlier years that have lower publication counts due to the increased difficulty in tracking down data from those years.” – It is not clear are the difficulties in tracking down data for the most recent years solved or not, and are the numbers of articles provided for the most recent years in the analysis accurate. For example, it is not explained why the number of articles in 2016 is significantly lower, compared with 2015.

Additional comments

This is improved version of the article, but the general impression is that experimental design has not improved significantly. The number of articles per year is not complete, and even though the authors state limitations in the collection of articles, it is unclear why the data for the latest years (except 2015) are not more reliable. The data on the share of articles already available in the open access is missing, and I strongly suggest the authors to add this, if not in this one, then in their future research. The authors' opinion is that usage rights and licensing are beyond the scope of the study, but this is an important issue for funders and should be considered as a part of the implemented policy. Open access is more complex than "gold" and "green" route.

---

## Round 0.3 · accepted · Accept

· Academic Editor

Accept

Thank you for answering the reviewers' comments.